# In Situ Simulation: A Strategy to Restore Patient Safety in Intensive Care Units after the COVID-19 Pandemic? Systematic Review

**DOI:** 10.3390/healthcare11020263

**Published:** 2023-01-14

**Authors:** Vanesa Gómez-Pérez, Dolores Escrivá Peiró, David Sancho-Cantus, Jorge Casaña Mohedo

**Affiliations:** 1Intensive Care Unit, La Fe Polytechnic and University Hospital, 46026 Valencia, Spain; 2Department of Nursing, Catholic University San Vicente Mártir, 46001 Valencia, Spain

**Keywords:** safety, patients, in situ simulation, intensive care units

## Abstract

Background: Patient safety is a public health problem worldwide. In situ simulation (ISS) arises as a learning strategy that allows health professionals to immerse themselves in a real environment without endangering the patients until they have learned the skills needed, thus increasing the quality of care. This systematic review aimed to verify the efficacy of the use of “in situ simulation” as a method that will allow health professionals to increase patient safety in Intensive Care Units after the situation experienced during the pandemic caused by the COVID-19 virus. Methods: Seven studies were reviewed using the PRISMA methodology for systematic reviews. The CASPe guide was used to assess the quality of the manuscripts. Results: The main topics that emerged from this review in relation to in situ simulation were as follows: looking at aspects such as patient self-perception of safety, adverse events, interprofessional communication and health system organization in relation to in situ simulation. Conclusions: The adequate implementation of in situ simulation after the COVID-19 pandemic in ICU services is shown to be an efficient and effective strategy to promote improvement in the attitudes on a culture of safety and teamwork of professionals.

## 1. Introduction

The report of the Committee on Quality of Care published in 2000 by the American Institute of Medicine [1], entitled To err is human: Building a safer health system, showed that healthcare is not as safe as it should be. It estimated that between 44,000 and 98,000 deaths per year were attributed to errors in health care. Eighteen months after, a second, more comprehensive report, Crossing the Quality Chasm [2], was published and served as a model for quality improvement and patient safety. Both reports suggested that healthcare professionals should adopt training methods currently used in military and commercial aviation, such as the use of simulation, to improve patient safety through the training of healthcare personnel. This would reduce medical errors, especially in areas involving non-technical skills such as teamwork, leadership and communication. These simulator-based teachings allow trainees to obtain a high level of training and practice in identifying and managing situations that could lead to a disaster, all without endangering any lives [3]. The World Health Organization launched the “Partnership for Patient Safety” [4] to promote actions, tools and recommendations to improve safety in all countries in the world. As early as 2006, the European Commission [5] urged countries to formulate policies, strategies and plans to improve patient safety in their medical institutions. In 2009, the Council of the European Union [6] issued the “Council Recommendations on Patient Safety,” including the prevention and control of healthcare-associated infections and emphasizing recommendations to promote education and training of healthcare workers.

The World Health Assembly discussed global action on patient safety in resolution WHA72.6 [7], and recognized the importance of “education, training and continuing professional development to create and maintain a competent, compassionate and committed health workforce operating in an environment conducive to safe health care.” According to data provided by the ENVIN Registry of the Spanish Society of Intensive Care Medicine, Critical Care and Coronary Units, which collects invasive device-related infections in critically ill patients, half of the patients with COVID admitted to the ICU had one or more types of respiratory infections, most of which were related to the equipment used in this setting. The incidence rates of mechanical ventilation-related pneumonias, primary catheter bacteremia and urethral catheter-related urinary tract infections, have increased two- to three-fold, and this has been accompanied by an increase in ICU length of stay and mortality [8].

The SYREC (Safety and Risk in Critical Patients) study [9] provided data on incidents and adverse events in intensive care medicine in Spain. Based on self-reported events, a prospective cohort study was conducted in 79 Spanish intensive care units. The median risk of suffering a non-harmful incident due to admission to the intensive care unit was 73% and that of suffering an adverse event was 40%. The most common adverse events were care-related and healthcare-associated infections. Ninety percent of non-harmful events and sixty percent of adverse events were classified as preventable or possibly preventable. Several studies [10,11] found that inexperienced healthcare personnel or those with few practical skills increased the risk of committing adverse events during clinical practice, which is why many studies endorse the prior training of professionals to bring them as close as possible to the reality of care, but without harming the patient, and encouraging critical reflection of the actions.

The current healthcare reality in Intensive Care Units (ICU) demands well-trained healthcare professionals with an adequate level of competence to provide quality care and an adequate culture of safety. There are multiple classifications of simulation in the literature, but among them, Jeffries and Cloches [12] classify it according to the technology utilized, into five types: hybrid simulation, new case simulation, standardized patients, in situ simulation and virtual simulation [13,14,15,16,17]. Within the existing classification of the different simulation modalities, the present focused on in situ simulation as a learning method and as a tool to recover patient safety in Intensive Care Units [18,19].

In situ simulation training (ISS) is a team-based training technique that takes place in patient care units, using the equipment and resources available in those units, and involving actual members of the healthcare team [3]. With ISS, it is possible to improve reliability and safety, especially in high-risk areas, such as the ICU, emergency and emergency departments [20]. It has been shown to have a positive effect on healthcare professionals’ reactions, changes in safety attitudes, organizational performance and teamwork. This type of training allows teams to review and reinforce their clinical problem-solving skills to prepare for a crisis or low frequency/high urgency events. It provides an approach to support and develop teamwork, leadership and communication skills, among others [21,22]. In general, international strategies in patient safety are mainly oriented into two major areas: cultural change in professionals, and implementation of safe practices, which includes the importance of professional training on which our review is based.

The objective of this review was to systematically assess the benefits of implementing “in situ” simulation as a strategy to improve patient safety in all care settings in intensive care units.

## 2. Materials and Methods

### 2.1. Search Strategy

The following search strategy was used to search each database. “patient*” OR “intensive care units” OR “in situ simulation” OR “safety”. This systematic review considered studies that evaluated in situ simulation in intensive care units and emergency departments, and those that explored patient safety. Different professional categories (physicians, nurses and auxiliary nursing care technicians) were included. Studies using other types of simulation than “in situ”, such as laboratory simulation, were excluded, as well as articles on pediatrics, neonates or others and studies from other areas of patient care.

In the first phase, after the search in the different databases was carried out independently by two researchers, a screening was performed in which the articles were included based on the review of the title. In a second phase, the articles were screened by two different researchers based on the review of the abstract according to the inclusion and exclusion criteria defined. Afterwards, the included articles were subjected to a critical reading using the PRISMA guidelines [23].

### 2.2. Elegibility Criteria

The bibliographic search was carried out using the snowball technique [24], and the following limitations were established: all studies published between 1 January 2011, and 31 December 2021, in English or Spanish, and type of scientific article; the databases used were Cochrane, Medline, CINAHL, Pubmed, Scopus, JBI and Web of Science. Table 1 shows the search strategy and the results obtained.

### 2.3. Assessment of Methodological Quality

Once the selection of the studies to be included in the systematic review had been made, the CASPe Template [25,26] was utilized to assess the quality of these articles. This tool is divided into three sections (validity, results and applicability). Disagreements that arose between reviewers were resolved through a discussion or with a third reviewer.

### 2.4. Data Collection and Synthesis

Data were extracted from the articles included in the review following the protocol established by the PRISMA group for systematic reviews [27]. The extracted data included specific aspects on interventions, populations, study methods, and main outcomes, and the quality of the articles was assessed with these data. A meta-analysis was not possible due to the great variability of data obtained in the search, so the results are presented in narrative form and illustrated with tables to synthesize the information.

## 3. Results

### 3.1. Overview of Studies

A total of 750 articles were identified after the search, of which 141 were excluded because they were duplicated in the different databases. Of the remaining 609 articles, 67 were selected after reading the title. Next, the abstract was read and 36 articles were excluded because they did not meet the inclusion criteria previously mentioned. Of the twenty-one potentially analyzable articles, a complete reading of the articles and the application of the established inclusion criteria were carried out to finally include seven articles for the present review (Figure 1).

The studies were conducted in France, Canada, Australia, Hong Kong, Switzerland, two of them in Brazil, Denmark and United Kingdom. Regarding the methodological design, ten of them were intervention studies, and one of them was a systematic review. It was decided to include it because of the relevance of the content. 42.8% (3/7) were pre-post-intervention studies, and the remaining 57.1% (4/7) were post-intervention. Two of them (28.5%) employed mixed quantitative and qualitative methodologies, using measurement scales in their results, while 71.4% of the studies were qualitative. A total of 36.36% of the studies were conducted in emergency departments, with the remaining 63.64% conducted in ICU departments. The topics analyzed in the review were adverse events, the perception of patient safety, communication between professionals and the organization of the health system (Table 2).

### 3.2. Detection of Latent Security Threats (LST)

Adverse events (AE) are often the result of several small errors. These errors are called latent safety threats because they occur prior to these events. LSTs are, by nature, difficult to predict or detect until a critical event occurs, often to the detriment of patient care. In situ simulation can identify these safety threats or precursors to AEs by reproducing clinical situations in a real-world setting, allowing LSTs to be addressed prospectively and proactively without risk to patients.

Through a case study, Bapteste [28] revealed that the mislabeling of medication on the crash cart led to an error in the administration of that medication and a delay in treatment, allowing for a correction of latent threats detected during ISS sessions. In a narrative review, Petrosoniak [29] revealed the identification and mitigation of medication-, equipment- or system-related LSTs detected through training with ISS, highlighting that it is more effective in the actual workplace than in the laboratory.

The prospective study conducted by Couto [30] allowed for a conclusion that training with ISS made it possible to detect a high rate of LST, among which we find those related to medication, absence of leader in the work group, and more frequently those related to equipment (difficulties with defibrillator, ventilator, material, vascular accesses).

### 3.3. Staff Perception of Patient Safety Culture

Through the Safety Attitude questionnaire, Schram [31] observed the proportion of participants with a positive attitude towards multidimensional aspects of patient safety before and after the on-site simulation sessions. Included were teamwork climate, safety climate, job satisfaction, stress recognition, working conditions, and leadership support for patient safety. The study showed an improvement in safety culture following the on-site simulation, by improving the proportion of staff with positive attitudes in five of the six dimensions of safety culture, thus highlighting greater effectiveness in the hospital performing acute care versus that performing care in an elective setting.

Similar findings were obtained by Paltved [32] when investigating whether in situ simulation improved patient safety attitudes. The results of his study concluded that the in situ simulation (ISS) program had a positive effect on the involved staff’s safety climate attitudes and teamwork climate after training, without obtaining a significant increase or decrease in the other categories.

Another study [33] found that participants improved their social skills on patient safety, thus improving the quality of patient care.

### 3.4. Interprofessional Communication

Inadequate communication can contribute to the occurrence of critical incidents (CI) and adverse events (AE). Miscommunication is relatively common and can appear directly as a medical error or be described as a contributing factor to the series of events that lead to a medical error. Therefore, Paltved [32] set out to investigate the intervention effects of ISS as a strategy to train interprofessional communication skills, through a needs analysis that informed the on-site simulation program, which included a thematic analysis of patient safety data and a short-term ethnographic study. The main outcome was the use of the Situation Background Assessment Recommendation communication tool, whereby staff rated communication in both information transfer and verbal order taking in a critical situation in order to improve patient safety. By solving problems that hindered communication skills such as interruptions that impaired communication, and the lack of organizational structures to support safe communication procedures, shared understanding and communication handoffs between professionals were improved, thereby reducing patient safety risk.

In turn, these data were supported by other studies [33] on in situ simulation, reflecting that SSI could be used as a tool to improve clinical decision making and interdisciplinary communication.

### 3.5. System Organization

In line with the above, Checuti [33] not only addressed human skills deficiencies, but also analyzed and addressed key environmental and organizational issues and exposed how these interacted and affected outcomes in patient care pathways. This enabled various components of the work system to be evaluated and modified, thereby increasing patient safety. He based this on the Systems Engineering Initiative model for patient safety, which is based on the understanding that errors in the patient care pathway are not due solely to individuals, but to the suboptimal systems with which they interact. This same model was also used by Andrew Petrosoniak [29] in his study to support the usefulness of SSI by providing information on system barriers and their corrections to provide high quality care. Such obstacles were also detected in the prospective study conducted by Couto [30] on LST related to systems and organization, which allowed for changes to be made in the service, such as location of essential material to facilitate access, acquisition of whiteboards to write down verbal orders in emergency situations, the placement of cognitive aids such as dosage tables of most commonly used medications, as well as updating of action protocols.

The aforementioned case study [28] concluded that the detection of latent threats proactively allowed not only the change in and review of labeling but also a change in the organization of the service, thus improving patient safety.

### 3.6. Assessment of Methodological Quality

Table 3 below shows the evaluation of each of the texts using the CASPE tool. As can be seen, most of the texts have obtained high scores, which means that they have a high quality (scores from 6 points).

## 4. Discussion

This systematic review confirms the implementation of the in situ simulation methodology as an effective way of addressing the elements that promote good patient safety, thus enabling the application of good practices during clinical care.

When discussing patient safety, we must address a set of elements that help us to minimize the risk of suffering an adverse event or mitigate its consequences in the health care process. These elements include achieving safe care processes by detecting latent threats that can prevent the occurrence of adverse events, maintaining professionals with a positive attitude towards the culture of safety in order to avoid failures in care, and managing or developing adequate interprofessional communication, among others. These elements are intertwined with each other to ensure quality and safety in care as stated by Kozlowski [38] in his research.

In a study [39] conducted in a Cincinnati children’s medical center, 64 scheduled SSI drills were conducted and 134 latent safety threats and knowledge gaps were identified and classified as threats to medication, equipment, or resources/systems, matching the same characteristics of LSTs found in two studies [29,30] analyzed in the present review. The identification of these errors resulted in the modification of systems to reduce the risk of error, with the study describing the usefulness of SSI to identify and resolve latent safety threats and improve the quality of care provided to pediatric patients. Another study conducted in 2012 at the US Pediatric Medical Hospital [40] analyzed 90 drills over a period of one year, in which a total of 218 professionals participated. Seventy-three LSTs were identified, a rate of one for every 1.2 simulations performed. Examples of the identified threats included equipment malfunctions and gaps in knowledge about role responsibilities. In contrast, we did not find any studies which did not describe the efficiency of SSI for the detection of LST, which demonstrates that the results provided by the scientific literature coincided with the findings of our review, and confirms the value of this methodology for increasing patient safety, which is in line with the study carried out by Fregene [37], who shows that ISS can detect these deficiencies and correcting them in debriefing.

Regarding the change in the attitude of the staff towards a culture of patient safety post-intervention with ISS sessions, the analysis of the studies included in the present review suggests a positive effect on the creation of an ISS program, showing a significant improvement in staff attitudes towards safety and teamwork climate. This was evidenced in the studies by Schram [31], Paltved [32] and Jonsson [34]. However, evidence supporting the effectiveness of intervention strategies to improve the culture of safety is limited in the literature, and results differ from each other, as noted in the systematic review by Weaver [41]. This significant increase in staff attitudes on teamwork climate is supported by studies such as Bleakley [42] and Cooper [43], conducted in the operating room area, where an improvement in teamwork climate was also found after the SSI intervention. However, Cooper [39], following an ISS-based anesthesia program, reported no effects on patient safety climate.

Both Paltved [32], Checuti [33], Eric [35] and Martins [36] agreed that ISS could be used as a tool to improve clinical decision making and interdisciplinary communication, as demonstrated by other studies found [44]. This latter study detected that the most common problem was communication, which correlates with the outcome of our results. We can thus state that SSI can be a method to improve patient safety due to the better opportunity provided for transferring good communication during the intervention in a real environment. However, Patterson [40] noted that after the SSI intervention, 77% of the providers of the post-intervention survey training found little or no clinical impact, even though the delivery of the SSI sessions were conducted for more than 1 year.

To optimize the effect of in situ simulation on patient outcomes, there is a need to move beyond the use of simulation merely as an educational intervention and human skills screening, to the integration of simulation as a patient safety tool at the organizational and system level. Similarly, this conclusion was supported by other authors [45] in a review conducted by a panel of five experts from a wide range of institutions, who discussed the implementation of simulation to improve systemic aspects of perioperative services, where simulation-based techniques were observed to be effective in conducting prospective root cause analyses to address system deficiencies leading to sentinel events. Therefore, this tool would help us to re-organize the affected services after the recent pandemic. We can find articles published in the last two years, such as the one by Laco [46], that demonstrate the use of SSI as a technique to reorganize services during the months of the pandemic.

Truchot [4], through a mixed method study to assess the feasibility and impact of in situ simulation, concluded that SSI offers the potential to improve patient safety through training, but may jeopardize the quality of the continuum of care by diverting human resources to the training process. To avoid this, he stresses the importance of planning and implementation strategies for simulation programs, since a poorly programmed exercise can have a double negative impact by failing to meet the proposed objectives. For this reason, several authors [29] also focused their studies on designing strategies for the good implementation of these training programs.

The study has some limitations. First of all, the number of articles found, and it is not known why there is such a small number of bibliographic references on this subject. A large number of articles were found on laboratory simulation, but fewer on simulation carried out in the real work site, such as in situ simulation, which may be due to the recent incorporation of this learning technique. On the other hand, most of the publications found on ISS were in the areas of anesthesia, obstetrics and operating rooms, services that were excluded in this review, as it focused on services that had been affected by the recent pandemic caused by COVID-19 such as the ICU and emergency and urgency services, and therefore, the literature found was scarce.

On the other hand, although the included articles were evaluated using the CASPe scale and showing their percentage in the results of the studies, it was observed that only four of them had an acceptable quality to be included in the systematic review, while the rest were of lower quality, as they were case studies in which there was a personal subjectivity that generated measurement and reporting biases. Some of them did not allow for comparisons to be made with another control group that did not participate in situ simulation training, and therefore they did not present solid evidence, although they may contribute to the improvement of quality and patient safety. These studies used a non-randomized sample, which conditions the results, as does the impossibility of using blinding in these investigations.

These were short-term studies where the results were analyzed once the simulations had been completed, and no studies were found where the long-term impact of the intervention carried out was evaluated.

### Implications for Practice

This systematic review suggests expanding lines of research by conducting longitudinal studies to investigate the impact of training and the evolution of safety attitudes that emerge over time. Research studies are needed on the effectiveness of ISS in ICU services, both in technical and non-technical skills, in order to increase the scientific evidence on this technique and to train health professionals with an adequate safety culture and provide high quality care.

## 5. Conclusions

The implementation of ISS training programs in the services affected by the COVID-19 pandemic would be an efficient and effective strategy to help solve the current problems found in patient safety. In order to be able to achieve our general objective, the basic elements that shape patient safety were analyzed in the different studies, concluding that ISS is a method that allows for the detection of latent threats. This makes it possible to avoid errors in medication, communication, and the system, but more frequently those related to equipment, thus showing its benefit in their detection and possible correction. On the other hand, it was observed how the implementation of ISS in intensive care and emergency departments improved communication both in the transfer of patients and in interprofessional communication in emergency situations, in addition to improving the attitude towards the culture of safety in the provision of care by these professionals. All of this allows for the evaluation and modification of components of the system by improving the systematic and organizational aspects of the services affected by the present pandemic.

## Figures and Tables

**Figure 1 healthcare-11-00263-f001:**
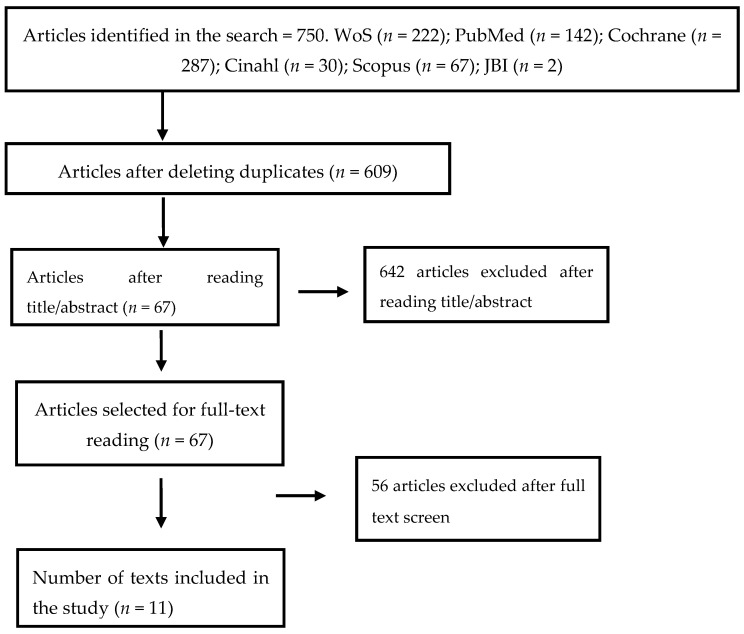
PRISMA flow diagram for study selection.

**Table 1 healthcare-11-00263-t001:** Search strategies in the different databases.

Search Strategies	Pubmed	Cochrane	WoS	Cinahl	Scopus	JBI
(((((((((patient) OR (patient *)) OR (client *)) OR (adult)) OR (aged)) AND (UCI)) OR (Intensive care units)) OR (ICU)) OR (emergencies)) AND (((simulation “in situ”) OR (high fidelity simulation training)) AND (patient safety)) Filters: in the last 10 years, Humans	142			30		2
patient in Title Abstract Keyword OR client in Title Abstract Keyword AND in situ simulation in Title Abstract Keyword AND intensive care units in Title Abstract Keyword OR emergencies in Title Abstract Keyword—with publication date in the Cochrane Library Between Jan 2011 and Jun 2021		287				
ALL = ((((((((((patient) OR (patient *)) OR (client *)) OR (adult)) OR (aged)) AND (UCI)) OR (Intensive care units)) OR (ICU)) OR (emergencies)) AND (((simulation “in situ”) OR (high fidelity simulation training)) AND (patient safety)))			222			
TITLE-ABS KEY (intensive AND care AND unit AND in AND situ AND simulation) AND (LIMIT-TO (PUBYEAR, 2021) OR LIMIT-TO (PUBYEAR, 2020) OR LIMIT-TO (PUBYEAR, 2019) OR LIMIT-TO (PUBYEAR, 2018) OR LIMIT-TO (PUBYEAR, 2017) OR LIMIT-TO (PUBYEAR, 2016) OR LIMIT-TO (PUBYEAR, 2015) OR LIMIT-TO (PUBYEAR, 2014) OR LIMIT-TO (PUBYEAR, 2013)) AND (LIMIT-TO (SUBJAREA, “MEDI”) OR LIMIT-TO (SUBJAREA, “NURS”)) AND (LIMIT-TO (LANGUAGE, “English”)) AND (LIMIT-TO (SRCTYPE, “j”))					67	

**Table 2 healthcare-11-00263-t002:** Effect of on-site simulation training.

Author/Study Design	Participants	Interventions	Outcome Measurements	Results	CASPe Score
Truchot et al. [4] Mixed method: qualitative method for the assessment of feasibility and acceptability and a quantitative method for the assessment of patient safety and participant risks.	Phase 1: Non-random, voluntary participation. (Announced or unannounced). Phase 2: random	40 min simulation sessions (20 min scenario and 20 min debriefing).	Semi-structured interviews were used to assess the acceptability of the intervention.	On-site simulation in an emergency department is feasible, safe and associated with benefits for both staff and patients.	9/11
Bapteste et al. [28] Case studies	New nurses and professionals	On site-simulation session in a room available in ICU. High fidelity Dummy (SimMan, Laerdal).	Participatory observational	Session 1: medication error. Session 2: delay in treatment.	6/11
Petrosoniak et al. [29] Systematic review and experimental review with case example	*n* = 117 manually reviewed papers	Cases with on-site simulation intervention. Group experience: 200 on-site simulation sessions in the emergency department in various countries	Bibliographic search in Pubmed, Medline, Scopus, Web of Science and ERIC.	Simulation training in the ED leads to tangible improvements in teamwork, safety and systems.	7/10
Couto et al. [30] Prospective Study	*n* = 114 participants in the scenarios. *n* = 101 in training tasks. *n* = 49 scenarios	Three scheduled 10-min on-site simulation scenarios alternated for each theme on a daily basis.	On-site simulation sessions followed by debriefing by two facilitators. Latent safety threats were identified using a checklist.	56 latent safety threats were detected, with an average of 1.1 per scenario.	10/11
Schram et al. [31] Cross-sectional pre-post intervention study	*n* = 967 healthcare professionals (39 trained as simulation instructors)	Interventions were conducted in the hospital setting (in situ), 54 sessions in Hospital 1 and 62 in Hospital 2. No systematic simulation was carried out prior to the intervention.	To measure outcomes, the Safety Attitudes Questionnaire (SAQ) was used, which investigates patient safety culture before the intervention and 4–8 weeks after the intervention.	The response rate varied between 63.6% and 72.0% between surveys and hospitals. Mean scores on the scale improved significantly in five of the six safety dimensions in hospital 1, while only one dimension improved significantly in hospital 2.	10/11
Paltved. et al. [32] Denmark Mixed qualitative (ethnography) and quantitative pre-post intervention study	*n* = 16 health teams composed by 9 doctors and 30 nurses.	Three-pronged strategy: 1. thematic analysis of patient safety data. 2. Needs analysis based on a short-term ethnography. 3. Pre-post assessment using the validated Safety Attitudes questionnaire.	A convergent parallel mixed method was used to collect both qualitative and quantitative data in parallel and the analysis was merged in the final phase.	The findings of this study suggested that an on-site simulation program can act as an important catalyst for the improvement of safety and teamwork attitudes.	9/11
Chetcuti and Bhowmick [33] Pre-post intervention study	*n* = 12-bed ICU hospital admitting 450 patients per year (Random selection).	On-site simulation sessions using the Laerdal SimMan EssentialTM dummy.	After the evaluation and treatment of each clinical case, a didactic report was carried out using the FAST-PAGE model. The recording of the sessions was used to facilitate the information.	The evaluation of the outcome through pre- and post-simulation questionnaires was positive, participants improved their human factor skills as well as confidence in handling critical situations.	8/11
Jonsson et al. [34] Switzerland Randomized controlled intervention study	*n* = 167 ICU nurses, distributed among 26 teams	1 control group and 1 intervention group are faced with an acute care situation to solve.	Evaluation through questionnaires and viewing of videos of the sessions.	Team leadership and task management improved in the intervention group	8/11
Eric et al. [35] Hong Kong Interdisciplinary group training programme	*n* = 1170 over 101 sessions	Groups made up of doctors, nurses and other health care professionals	35-item questionnaires and a 13-item questionnaire related to the quality of training.	Simulation-based training contributed significantly to preparing hospital staff, reinforcing protocols and workflow for endotracheal intubation.	10/11
Martins et al. [36] Brazil Pre-post test design for simulation training	*n*= 48 doctors, nurses and nursing technicians	Pre- and post-simulation study	Knowledge test	Simulation equips professionals with skills to deal with COVID-19, generating benefits for health systems, professionals and patients.	7/11
Fregene et al. [37] London Pre-post test design for simulation training	*n*= 32 from the departments of anesthesia	A total of 8 scenarios were carried out	Corrective measures are established for errors detected during the simulation.	It showed that on-site simulations identified multiple operational deficiencies in the ICU isolation room and allowed corrective action to be taken prior to admission of the first patient with COVID-19.	8/11

**Table 3 healthcare-11-00263-t003:** Quality criteria according to the CASPe scale.

Question	Truchot [4]	Bapteste [28]	Petrosoniak [29] *	Couto [30]	Schram [31]	Paltved [32]	Chetcuti [33]	Jonsson [34]	Eric [35]	Martins [36]	Fregene [37]
Q1	1	1	1	1	1	1	1	1	1	1	1
Q2	1	0	1	1	1	1	1	0	1	1	1
Q3	1	0	1	1	1	1	0	1	0	1	1
Q4	1	0	0	1	1	1	1	1	1	0	0
Q5	0	0	1	0	0	0	0	0	1	0	1
Q6	Concluding	Concluding	Concluding	Concluding	Concluding	Concluding	Concluding	Concluding	Concluding	Concluding	Concluding
Q7	P.A. **	P.A. **	P.A. **	G.A. ***	G.A. ***	P.A. **	P.A. **	P.A. **	G.A. ***	P.A. **	P.A. **
Q8	1	1	1	1	1	1	1	1	1	1	0
Q9	1	1	0	1	1	1	1	1	1	1	1
Q10	1	1	1	1	1	1	1	1	1	1	1
Q11	1	1	X	1	1	1	1	1	1	0	1
CASPe score	9/11	6/11	7/10	10/11	10/11	9/11	8/11	8/11	10/11	7/11	8/11

* The CASPe scale model for reviews is used, which has 10 items. ** Poor Accuracy. *** Good Accuracy.

## Data Availability

Not applicable.

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
