# Peer review of "In Situ Simulation: A Strategy to Restore Patient Safety in Intensive Care Units after the COVID-19 Pandemic? Systematic Review"

_healthcare, 2023, doi:10.3390/healthcare11020263_

Round 1

Reviewer 1 Report

Healthcare 2143408

This systematic review highlighted the importance of in situ simulation on improving patients’ outcome in ICU. However, the overall quality of this review is compromised due to the major flaw of search strategies and quality appraisal. 

1. Search terms, the authors are highly encouraged to consult a librarian to refine the search strategies. 

Several key terms and databases were ignored, i.e., mesh term “intensive care units” (https://www.ncbi.nlm.nih.gov/mesh/68007362) and “simulation”.

Here is the difference between Medline and PubMed. https://www.nlm.nih.gov/bsd/pmresources.html

The JBI and Scopus should be included.

It would be better to describe each category of the search terms, such as which words/mesh terms were used for patients, which were used for ICU, and which were used for simulation.

A detailed search results should be attached.

The current table 1 shows no record yielded in WOS, but figure 1 shows 691 records emerged. The figure 1 needs more work. Please be kind to verify the numbers.

2. quality appraisal, please be kind to add more details regarding the evaluation criteria, an overall CASPe score may cover the quality of included studies, i.e., readers need more information to distinguish the difference between studies scored 6/11 and 10/11.

Author Response

Point 1: This systematic review highlighted the importance of in situ simulation on improving patients’ outcome in ICU. However, the overall quality of this review is compromised due to the major flaw of search strategies and quality appraisal. 

  1. Search terms, the authors are highly encouraged to consult a librarian to refine the search strategies. 

Several key terms and databases were ignored, i.e., mesh term “intensive care units” (https://www.ncbi.nlm.nih.gov/mesh/68007362) and “simulation”.

Response 1: The reviewer's comments are appreciated. The search included terms such as "intensive care unit", as shown in Table 1, although it is true that this is not specified clearly enough in the text.

Point 2: Here is the difference between Medline and PubMed. https://www.nlm.nih.gov/bsd/pmresources.html

Response 2: Rectified and adapted

Point 3: The JBI and Scopus should be included.

Response 3: Ot The following text is included in the limitations section, since it is not possible to carry out a new search at this time; the comment is appreciated and is valued as a furure proposal. Other databases such as JBI or Scopus could have been used to refine the search, so an extension of the present study is proposed for the future.

Point 4: It would be better to describe each category of the search terms, such as which words/mesh terms were used for patients, which were used for ICU, and which were used for simulation.

Response 4: Modified according to the reviewer's indications

Point 5: A detailed search results should be attached.

Response 5: The search equations are reviewed and the results obtained are refined in response to the reviewer's suggestion.

Point 6: The current table 1 shows no record yielded in WOS, but figure 1 shows 691 records emerged. The figure 1 needs more work. Please be kind to verify the numbers.

Response 6: Figure 1 and Table 1 are revised and the original searches are re-run, adjusting for errors.

Point 7: quality appraisal, please be kind to add more details regarding the evaluation criteria, an overall CASPe score may cover the quality of included studies, i.e., readers need more information to distinguish the difference between studies scored 6/11 and 10/11.

Response 7: This tool is divided into three sections (validity, results and applicability). Each of the questions can be answered as "yes", "no" or "don't know". A proportion is made with the affirmative answers in relation to the number of questions to establish the systematic quality of the documents under study in this review, and the higher the weighting, the better the quality of the article reviewed

Reviewer 2 Report

In this manuscript authors present a review of literature about in situ simulaiton training and its possible role in improving patient safety in intensive care units. The rationale for this study is well explained in Introduction section. Methods are well described and discussion follows the results. There are some things I would like the authors to adress:

1.       In Matherials and methods section, row 101-102, authors should specify how was the screening of the articles in the second phase done, by one author or by two authors independently?

2.       In the Results section, authors say that all 7 studies were intervention studies. However, study by Petrosoniak et al. is a systematic review but still is analyzed in this manuscript?

3.       In the Reuslts section, row 174, a term SSI program appears. However, there is no explanation of the abbreviation.

4.       In the Results section, row 194, a reference number is written in superscript.

Author Response

In this manuscript authors present a review of literature about in situ simulaiton training and its possible role in improving patient safety in intensive care units. The rationale for this study is well explained in Introduction section. Methods are well described and discussion follows the results. There are some things I would like the authors to adress:

Point 1: In Matherials and methods section, row 101-102, authors should specify how was the screening of the articles in the second phase done, by one author or by two authors independently?

Response 1: Thank you very much for your appreciation, it is specified in the corresponding section that in the second phase two authors intervened to revise the texts independently.

 Point 2: In the Results section, authors say that all 7 studies were intervention studies. However, study by Petrosoniak et al. is a systematic review but still is analyzed in this manuscript?

Response 2: The following content is specified: 6 of them were intervention studies, and one of them was a systematic review. It was decided to include it because of the relevance of the content.

Point 3: In the Reuslts section, row 174, a term SSI program appears. However, there is no explanation of the abbreviation.

 Response 3: Revised and modified

 Point 4: In the Results section, row 194, a reference number is written in superscript.

Response 4: Revised and modified

Round 2

Reviewer 1 Report

A review would not be a systematic review if the database JBI and Scopus was not included in the search strategies. Please be kind to get more information regarding the definitions of systematic review, scope review, and literature review. The authors insisted to not search in the mentioned databases, other than stated it as a limitation. In this case, the title should be revised. Then, there is limited meaning to publish this manuscript. The authors may request a longer time for revision.

It would be better to provide detailed information regarding the CASPe score in each of the 11 domains, i.e., "yes", "no" or "don't know" on every single domain evaluated for the included studies. Though studies rated 10/11 or 6/11, they may have different scores on a single item.

Author Response

Dear editors and reviewers, we are attaching a cover letter following your indications and hope that the explanations are satisfactory. We look forward to hearing from you.

Yours sincerely,

Reviewer 1

Point 1: A review would not be a systematic review if the database JBI and Scopus was not included in the search strategies. Please be kind to get more information regarding the definitions of systematic review, scope review, and literature review. The authors insisted to not search in the mentioned databases, other than stated it as a limitation. In this case, the title should be revised. Then, there is limited meaning to publish this manuscript. The authors may request a longer time for revision.

Response 1: In response to the reviewer's comment, a new search has been performed including the two suggested databases (Scopus and JBI). As a consequence of the results obtained, aspects of the manuscript have been modified, such as the flow diagram, results and discussion.

Point 2: It would be better to provide detailed information regarding the CASPe score in each of the 11 domains, i.e., "yes", "no" or "don't know" on every single domain evaluated for the included studies. Though studies rated 10/11 or 6/11, they may have different scores on a single item.

Response 2: In response to the reviewer's comment, a new table specifying the detailed scores for each of the texts has been added and further explained in the manuscript.